# Shifted inverse stereographic normal distributions as a flexible distribution family on the hypertorus

## Abstract

Circular data arises in various fields including robotics, biology, geology and material sciences. Modelling such data requires flexible distribution families on the hypertorus. Common choices are the von Mises and the wrapped normal distributions. In this work we investigate the *inverse stereographic normal distribution* as an interesting and computationally appealing alternative. We demonstrate its flexibility and practical applicability by fitting mixtures of shifted inverse stereographic normal distributions via gradient descent to dihedral data of protein backbones characterizing the conformational landscape of folding. Furthermore, we prove that the inverse stereographic normal distribution is unimodal if and only if all eigenvalues of the covariance matrix are less than or equal to 0.5.

## 1 Introduction

Many relevant problems in the fields of biology, geology, material sciences, robotics and engineering involve the study of circular random variables $\alpha_1, \ldots, \alpha_n \in [0, 2\pi[$. Prominent examples are the torsion angles of a protein backbone (Boomsma et al. (2008),Mardia et al. (2012)) or the angles describing the kinematics of a robotic arm (e.g. see Denavit & Hartenberg (1955)). Formally, the domain of such data is referred to as the *hypertorus*

$$\mathbb{T}_n := \underbrace{\mathbb{S}^1 \times \cdots \times \mathbb{S}^1}_{n},$$

where $\mathbb{S}^1$ is the unit circle. The field of directional statistics provides a range of distribution families on $\mathbb{T}_n$ (see Mardia & Jupp (2000) for an overview, or Ley & Verdebout (2018), Pewsey & García-Portugués (2020) for more recent references). Note that we can identify $\mathbb{T}_n$ by $[-\pi, \pi[^n$ via polar coordinates. Reasonable choices of modelling distributions should respect the topology of $\mathbb{T}_n$ in the sense that they satisfy periodic boundary conditions on $[-\pi, \pi[^n$.

Of specific interest are distributions that constitute a toroidal analogue to the normal distribution, given its favourable properties such as flexibility, "universal density approximation property" of Gaussian mixture models (see Nguyen et al. (2020) Theorem 5), limit distribution in Central limit theorem, among others. In this context, the *von Mises distribution* (first introduced by von Mises (1918)) and the *wrapped normal distribution* (e.g. see Mardia & Jupp (2000) page 50-51) are commonly suggested.

The von Mises density in one dimension is given by

$$
\begin{aligned}
f_{\mu,\kappa} : [-\pi, \pi[ &\longrightarrow \mathbb{R}_+ \\
\alpha &\mapsto \frac{1}{2\pi I_0(\kappa)} \exp\left(\kappa \cos\left(\alpha - \mu\right)\right),
\end{aligned}
\tag{1}
$$

where $\mu \in [-\pi, \pi[$ is called *mean direction*, $\kappa \geq 0$ the *concentration* and $I_0(\kappa) = \frac{1}{2\pi} \int_0^{2\pi} e^{\kappa \cos(\alpha)} \, d\alpha$ is the modified Bessel function of the first kind and order 0. Generalizations of the von Mises distribution to the bivariate case (Mardia (1975)) and higher dimensions (Mardia et al. (2008),Navarro et al. (2017)) have

been investigated in literature, however a major drawback for many applications remains the intractable normalization constant. For the one dimensional case, it was shown in Kent (1978) that the von Mises distribution closely approximates the wrapped normal distribution (see Collett & Lewis (1981), Pewsey & Jones (2005) for statistical considerations about discrimination of both distributions).

In general a probability density $p_w$ on $[-\pi, \pi[^n$ can be constructed from a probability density $p : \mathbb{R}^n \longrightarrow \mathbb{R}_+$ by "wrapping" it around the hypertorus in the following sense

$$
\begin{aligned}
p_w : [-\pi, \pi[^n &\longrightarrow \mathbb{R}_+ \\
\alpha &\mapsto \sum_{\underline{j} \in \mathbb{Z}^n} p(\alpha + 2\pi \underline{j}).
\end{aligned}
$$

In the case of $p$ being the density of a normal distribution with mean $\mu \in \mathbb{R}^n$ and covariance matrix $\Sigma \in \mathbb{R}^{n \times n}$, we obtain the wrapped normal density

$$
\begin{aligned}
p_{\mu,\Sigma}^{wn} : [-\pi, \pi[^n &\longrightarrow \mathbb{R}_+ \\
\alpha &\mapsto (2\pi)^{-\frac{n}{2}} \det(\Sigma)^{-\frac{1}{2}} \sum_{\underline{j} \in \mathbb{Z}^n} e^{-\frac{1}{2}\left(\alpha - \mu + 2\pi \underline{j}\right)^T \Sigma^{-1}\left(\alpha - \mu + 2\pi \underline{j}\right)}.
\end{aligned}
$$

In any practical context the infinite sum is truncated by introducing some $J \in \mathbb{N}$ and summing over the set $\mathcal{J} := \{-J, \cdots, J\}^n$ instead:

$$
\begin{aligned}
\hat{p}_{\mu,\Sigma}^{wn} : [-\pi, \pi[^n &\longrightarrow \mathbb{R}_+ \\
\alpha &\mapsto (2\pi)^{-\frac{n}{2}} \det(\Sigma)^{-\frac{1}{2}} \sum_{\underline{j} \in \mathcal{J}} e^{-\frac{1}{2}\left(\alpha - \mu + 2\pi \underline{j}\right)^T \Sigma^{-1}\left(\alpha - \mu + 2\pi \underline{j}\right)}.
\end{aligned}
$$

One drawback of the truncated wrapped normal distribution is the exponentially increasing computational complexity in the number of dimensions $n$ (since the number of terms in the sum is $(2J+1)^n$).

Selvitella (2019) suggested the inverse stereographic projection of normal distributions, termed *inverse stereographic normal distributions*, as a further flexible alternative to the von Mises distribution. Beyond having an easy tractable density in the multivariate case, the distribution was shown in Selvitella (2019) to have many favorable statistical properties. Specifically, the inverse stereographic normal distribution is closed under marginalization and conditioning, has asymptotic relations to the von Mises and wrapped normal distributions and is the limit distribution in a toroidal analogue of the central limit theorem. Furthermore Selvitella (2019) stated unimodality conditions for the inverse stereographic normal distributions in the one dimensional case and demonstrated applications in one and two dimensions.

In this work we consider shifted versions of the inverse stereographic projection of zero centered normal distributions, which we term *shifted inverse stereographic normal distributions* in reference to Selvitella (2019). We demonstrate the flexibility and practical applicability of the distribution family by fitting mixtures of shifted inverse stereographic normal distributions to non-trivial toroidal distributions in higher dimensions. Furthermore we generalize the unimodality result of Selvitella (2019) for the mean-free case to arbitrary dimensions.

We structured the article as follows: In section 2 we first formally introduce the inverse stereographic normal distribution. We then present our main theoretical result identifying the parameter set of unimodality for the inverse stereographic distribution. Finally we outline how mixtures of shifted inverse stereographic normal distributions can be fitted by gradient descent, restricting learning via diagonal parametrization to the subset of unimodality. Initial mean parameters are set by applying a clustering algorithm. In section 3 we fit mixtures of inverse stereographic distributions to several data sets, including wind direction data, samples from a special case of the bivariate von Mises distribution and torsion angle data of protein conformational landscapes. Specifically, we quantitatively investigate the distribution learning for alanine tetrapeptide (6 torsion angles) and chignolin (18 torsion angles). The quality of the fit is evaluated in terms of the estimated Kullback-Leibler divergence and visualized in TICA plots. We summarize our results and provide an outlook for future work in section 4.

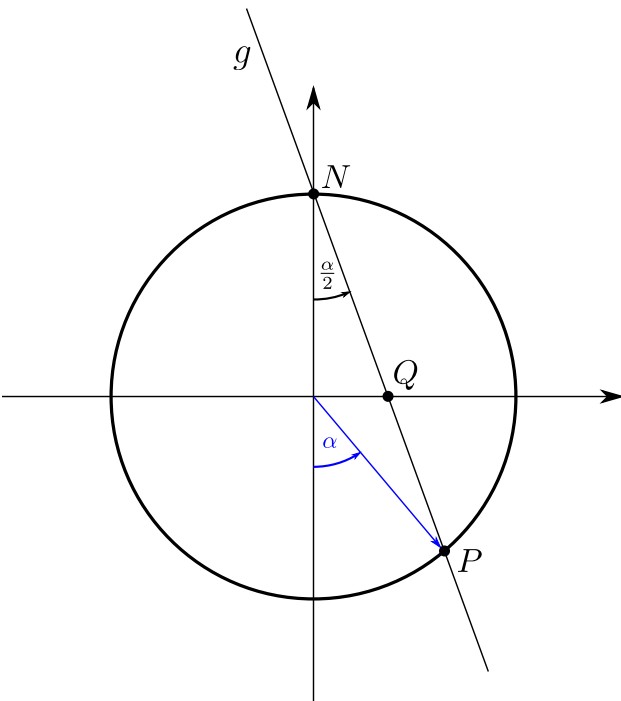

Figure 1: The point $P \neq N$ on the circle (parametrized by angle $\alpha \in ]-\pi, \pi[$) is mapped to the point $Q$ on the real line via the stereographic projection: The point $Q$ is defined as the intersection of the real line with the line crossing the pole $N$ and point $P$. Note that the coordinates of $Q$ are $Q = \left(0, \tan\left(\frac{\alpha}{2}\right)\right)$.

## 2 Methods

### 2.1 Stereographic projection

The stereographic projection is a $C_1$-diffeomorphism from the unit circle (except for the pole) onto the real line. For the unit circle being parametrized by an angle $\alpha \in ]-\pi, \pi[$, it is defined as

$$
\begin{aligned}
h: \quad ]-\pi, \pi[ \quad &\longrightarrow \quad \mathbb{R} \\
\alpha \quad &\mapsto \quad \tan\left(\frac{\alpha}{2}\right)
\end{aligned} \tag{2}
$$

Geometrically, a point from the circle $P \in S_1 \setminus \{(0,1)\}$, parametrized by an angle $\alpha \in ]-\pi, \pi[$, is mapped to a point $Q$ on the real line as follows: The point $Q$ is defined as the intersection of the straight line $g$ through $(0,1)$ and $P$ with the x-axis (see Figure 1).

### 2.2 Inverse stereographic normal distribution

By applying the mapping 2 component-wise, we obtain

$$
\begin{aligned}
h_n: \quad ]-\pi, \pi[^n \quad &\longrightarrow \quad \mathbb{R}^n \\
(\alpha_1, \ldots, \alpha_n) \quad &\mapsto \quad \left(\tan\left(\frac{\alpha_1}{2}\right), \ldots, \tan\left(\frac{\alpha_n}{2}\right)\right).
\end{aligned}
$$

Note that $h_n$ is a $C_1$-diffeomorphism, with functional determinant given by

$$\det\left(D_{h_n}(\alpha_1,\ldots,\alpha_n)\right) = \prod_{i=1}^{n} \frac{1}{1+\cos\left(\alpha_i\right)}.$$

Let $\mathcal{N}(0,\Sigma)$ be the centered normal distribution with covariance matrix $\Sigma \in \mathbb{R}^{n\times n}$. Using the change of variables theorem, we find the pushforward measure of $\mathcal{N}(0,\Sigma)$ under $h_n^{-1}$ to be

$$\mathcal{N}(0,\Sigma) \circ h_n = C \prod_{i=1}^{n} \left(\frac{1}{1+\cos(\alpha_i)}\right) e^{-\frac{1}{2} h_n(\alpha_1,\cdots,\alpha_n)^T \Sigma^{-1} h_n(\alpha_1,\cdots,\alpha_n)} d\alpha_1 \cdots d\alpha_n, \tag{3}$$

where

$$C = (2\pi)^{-\frac{n}{2}} \det\left(\Sigma\right)^{-\frac{1}{2}}$$

is the normalization constant. In other words, mapping a $\mathcal{N}(0,\Sigma)$ distributed random variable to $]-\pi,\pi[^n$ via $h_n^{-1}$ results in a random variable distributed according to (3). We call the density in 3 the *inverse stereographic normal density.*

We now define the *shifted inverse stereographic normal density* (SISND) by continuously extending the density 3 to $[-\pi,\pi[^n$ and subtracting a shifting parameter in its argument.

**Definition 1 (Shifted inverse stereographic normal density).** *For $n \in \mathbb{N}$ let*

$$S_+^n := \left\{X \in \mathbb{R}^{n\times n}|X = X^T, X \succeq 0\right\}$$

*be the set of symmetric, positive semidefinite matrices in $n$ dimensions. We denote the component-wise application of a function with an underscore as specified in Notation 1. For a covariance matrix $\Sigma \in S_+^n$ consider the following function*

$$
\begin{aligned}
g_\Sigma : \mathbb{R}^n &\longrightarrow \mathbb{R}_+ \\
\alpha = (\alpha_1,\cdots,\alpha_n) &\mapsto 
\begin{cases}
\prod_{i=1}^{n}\left(\frac{1}{1+\cos(\alpha_i)}\right) e^{-\frac{1}{2}\underline{\tan}\left(\frac{\alpha}{2}\right)^T \Sigma^{-1} \underline{\tan}\left(\frac{\alpha}{2}\right)}, & \alpha \in \mathbb{R}^n \setminus \{\pi + 2\pi k \,|\, k \in \mathbb{N}\} \\
0, & \alpha \in \{\pi + 2\pi k \,|\, k \in \mathbb{N}\}
\end{cases}.
\end{aligned}
$$

*Note that $g_\Sigma$ is continuous and periodic. For a covariance matrix $\Sigma \in S_+^n$ we define the shifted inverse stereographic normal density of center $\mu \in [-\pi,\pi[^n$ as*

$$
\begin{aligned}
f_{\Sigma,\mu} : [-\pi,\pi[^n &\longrightarrow \mathbb{R}_+ \\
\alpha = (\alpha_1,\cdots,\alpha_n) &\mapsto C \cdot g_\Sigma(\alpha - \mu), \tag{4}
\end{aligned}
$$

*where*

$$C = (2\pi)^{-\frac{n}{2}} \det\left(\Sigma\right)^{-\frac{1}{2}}.$$

*is the normalization constant. For the sake of brevity, we refer to the density in equation 4 by the acronym SISND (**s**hifted **i**nverse **s**tereographic **n**ormal **d**ensity).*

Depending on the covariance matrix $\Sigma \in \mathbb{R}^{n\times n}$ the density $f_{\Sigma,\mu}$ in 4 can be either unimodal or multimodal (see Figure 2). In fact, the eigenvalues of $\Sigma$ determine the modality, as specified in the following theorem:

**Theorem 1.** *Let*

$$\mathcal{A} := \left\{\Sigma \in S_+^n |\, \lambda_{max}\left(\Sigma\right) \le 0.5\right\} \tag{5}$$

*be the set of SPD matrices with all eigenvalues being less or equal than 0.5. For $\Sigma \in S_+^n, \mu \in [-\pi,\pi[^n$ let $f_{\Sigma,\mu}$ be defined as in Definition 1. Then $f_{\Sigma,\mu}$ is unimodal if and only if $\Sigma \in \mathcal{A}$.*

*Proof.* See Appendix F $\qquad\qquad\square$

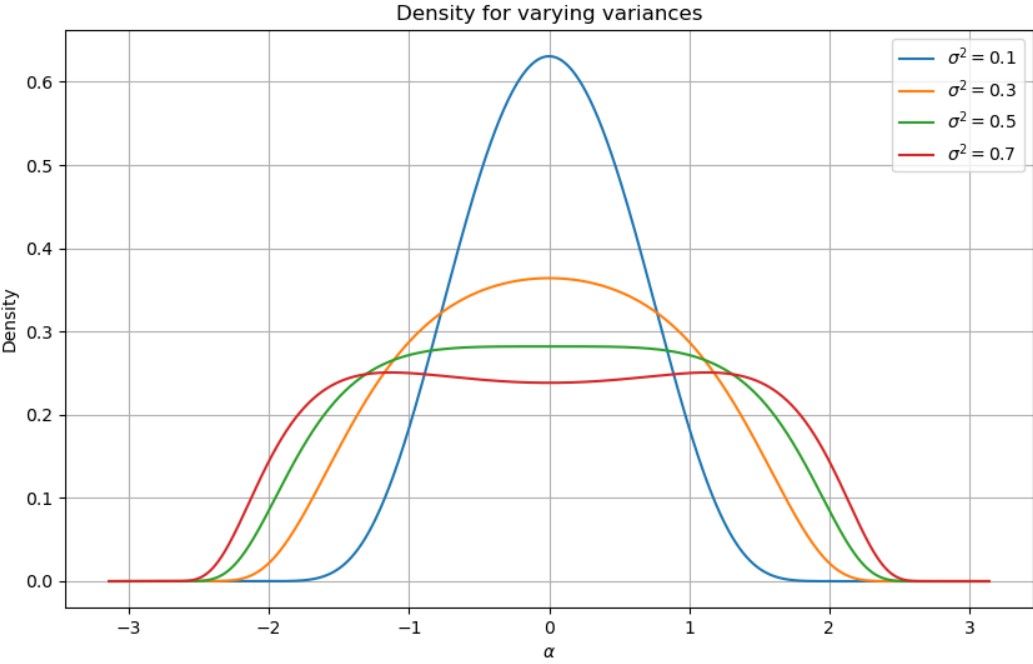

Figure 2: The inverse stereographic normal density as introduced in Definition 1 can be either unimodal or multimodal depending on the eigenvalues of the covariance matrix $\Sigma$. In this visualization we denoted $\Sigma = \sigma^2$. The density is unimodal if and only if $\sigma^2 \leq 0.5$.

### 2.3 Fitting mixtures of SISND

Let $m \in \mathbb{N}$ be the number of mixture components. The parameter space is given by

$$\Theta = \left\{ (w_i, \Sigma_i, \mu_i)_{i=1}^m \in \left( \mathbb{R}_+ \times \mathbb{R}^{n \times n} \times \mathbb{R}^n \right)^m \mid w_1 + \cdots + w_m = 1 \right\}.$$

For $\theta = (w_i, \Sigma_i, \mu_i)_{i=1}^m \in \Theta$ we define the mixture density

$$p_\theta = \sum_{i=1}^m w_i \cdot f_{\Sigma_i, \mu_i}.$$

Given samples $x_1, \ldots, x_k \in [-\pi, \pi[^n, k \in \mathbb{N}$, the maximum likelihood estimate

$$\theta_{ML} := \arg\max_{\theta \in \Theta} \prod_{i=1}^k p_\theta(x_i) = \arg\max_{\theta \in \Theta} \sum_{i=1}^k \log\left(p_\theta(x_i)\right)$$

is approximated via gradient descent. We initialize the means $(\mu_i)_{i=1}^m$, by first performing k-means clustering on the embedding of the toroidal data into a higher dimensional space. Let

$$\begin{aligned} \phi: \quad [-\pi, \pi[^n \quad &\longrightarrow \quad \mathbb{R}^{2n} \\ (\alpha_1, \cdots, \alpha_n) \quad &\mapsto \quad (\cos(\alpha_1), \sin(\alpha_1), \ldots, \cos(\alpha_n), \sin(\alpha_n)) \, . \end{aligned} \tag{6}$$

be the embedding of the $n$-dimensional toroidal data into $\mathbb{R}^{2n}$. Applying k-means clustering we obtain the cluster centers $c_1, \ldots, c_k \in \mathbb{R}^{2n}$. We normalize $c_1, \ldots, c_k$ to be on the torus and transform back to the angular representation: For $i \in \{1, \ldots, n\}$ and $c_i = (c_{i,1}, \ldots, c_{i,2n})$, define the initial centers

$$\mu_i := \phi^{-1}\left(\left(\frac{c_{i,1}}{\sqrt{c_{i,1}^2 + c_{i,2}^2}}, \frac{c_{i,2}}{\sqrt{c_{i,1}^2 + c_{i,2}^2}}, \ldots, \frac{c_{i,2n-1}}{\sqrt{c_{i,2n-1}^2 + c_{i,2n}^2}}, \frac{c_{i,2n}}{\sqrt{c_{i,2n-1}^2 + c_{i,2n}^2}}\right)\right).$$

Empirically we found it beneficial to restrict the learning to the set of unimodality $\mathcal{A}$ (see 5) of covariance matrices of eigenvalues less or equal than $\frac{1}{2}$. To this end, we parametrize the covariance matrices by a diagonal decomposition

$$\Sigma = Q\Lambda Q^T,$$

where $Q \in SO(n)$ and $\Lambda \in \mathbb{R}^{n \times n}$ is diagonal, s.th. $0 \leq \Lambda_{i,i} \leq 0.5$ for $i \in \{1, \ldots, n\}$. The special orthogonal group $SO(n)$ can be parameterized via the set of skew symmetric matrices $\mathfrak{so}(n) := \{A \in \mathbb{R}^{n \times n} \,|\, A^T = -A\}$, since the map

$$\begin{aligned} \psi : \mathfrak{so}(n) &\longrightarrow SO(n) \\ A &\mapsto \exp(A) \end{aligned} \tag{7}$$

is surjective (see Theorem 18.1 in Gallier (2011) for a proof). We note that for small eigenvalues the SISND approximates the wrapped normal distribution (see Figure 7 and compare Selvitella (2019)).

## 3 Experiments

In this section we demonstrate the flexibility of the suggested model by applying it to several examples. We start with two rather simple target distributions in 3 and 2 dimensions (wind direction data and a special case of the bivariate von Mise distribution) and then consider two more complex distributions in 6 and 18 dimensions (the backbone torsions of alanine tetrapeptide and chignolin). For the latter two the distribution fit is quantitatively evaluated by comparison against a baseline model.

### 3.1 Wind direction data

We first consider a small trivariate data set consisting of $1,682$ observations of wind directions available as dataset "WindDirectionsTrivariate" from the R package CircNNTSR (see Fernández-Durán & Gregorio-Domínguez (2016)). The measurements were taken at three different locations (San Agustin in the north, Pedregal in the southwest, and Hangares in the southeast) of the Mexico valley between January 1, 1993 and February 29, 2000. We display the 2 dimensional marginal distributions of the data as well as a fitted mixture model consisting of 50 components of SISND in Figure 8 of Appendix A.

### 3.2 The Sine model

The *Sine model* was introduced in Mardia et al. (2007) as a special case of the bivariate von Mises distribution with analytically known normalization constant. The density function is given by

$$f_S(\alpha_1, \alpha_2) = \frac{1}{C} \exp\left(\kappa_1 \cos(\alpha_1 - \mu_1) + \kappa_2 \cos(\alpha_2 - \mu_2) + \lambda \sin(\alpha_1 - \mu_1)\sin(\alpha_2 - \mu_2)\right),$$

where

$$C = 4\pi^2 \sum_{m=0}^{\infty} \binom{2m}{m}\left(\frac{\lambda^2}{4\kappa_1\kappa_2}\right)^m I_m(\kappa_1)I_m(\kappa_2),$$

and $I_m$ is the modified Bessel function of the first kind and order $m \in \mathbb{N}$. We chose $\kappa_1 = 0.8, \kappa_2 = 1.5, \mu_1 = \mu_2 = 3.0, \lambda = 2.5$, drew $1,000,000$ samples and fitted a mixture of SISND mixture of 5 components. The density plots are shown in Figure 9 of Appendix A.

### 3.3 Protein backbone data

In the following we analyze the fitting performance of mixtures of SISND for a varying number of components. A mixture model consisting of independent joint distributions of von Mises distributions serves as a baseline (compare Appendix C for a formal definition). We consider two different target distributions of torsion angles of protein backbones.

The first example is the distribution of 6 torsion angles determining the backbone conformations of alanine tetrapeptide. We generated the data by classical MD simulations at a temperature of 300 K with 2,000,000 iterations.

The second example is chignolin, a protein consisting of 10 amino acids, the backbone conformation can be expressed in terms of 18 torsion angles. The data consists of 2,000,000 samples and was obtained from Culubret & Fabritiis (2021) and generated using extensive MD simulations at 350 K (see Culubret & Fabritiis (2021) for more algorithmic and parameter details).

We evaluate the quality of the model fit in terms of the estimated Kullback-Leibler divergence. Since the underlying probability density $p$ is unknown, we approximate the Kullback-Leibler divergence $D_{KL}(p\|q)$ by the estimator introduced in equation (11) of Definition 2 in the Appendix.

Mixture models of increasing number of components are fitted by gradient descent optimization to approximate the empirical distribution of the torsion angles. Training was performed on a GPU of 24 GB memory (GeForce RTX 3090) with a batch size of 40000 for 150 epochs.

In Figure 3 we compare on alanine tetrapeptide the estimated KL divergence for mixture models of an increasing number of components consisting once of SISND and once of independent joint distributions of von Mises distributions. We observe that the SISND mixture model outperforms the baseline and converges substantially faster. For more than 100 components the SISND mixture seems to saturate, while the baseline is not reaching that value even for 300 components.

Similarly, the Figure 4 displays the divergence results for chignolin. The evaluation measure indicates that the fitting performance might still further improve for both models taking more than 300 components, reflecting the complexity of the target distribution. The mixture of SISND again significantly outperforms the baseline for all numbers of components.

Finally, we visualize the distribution fit of the SISND mixture of 300 components by TICA plots. TICA (**T**ime-lagged **I**ndependent **C**omponent **A**nalysis) was first suggested by Molgedey & Schuster (1994) and introduced into the field of molecular dynamics and computational chemistry by Pérez-Hernández et al. (2013) and Schwantes & Pande (2013). Similarly to PCA, TICA provides an orthonormal basis of the $n \in \mathbb{N}$ dimensional vector space, and the projection onto the subspace spanned by the first $\mathbb{N} \ni m \leq n$ basis vectors constitutes a dimensionality reduction technique. However, in contrast to PCA, TICA requires time series data and the basis vectors are chosen as directions of maximal autocorrelation of the time series. The latter is usually termed as TICA defining the successive subspaces of "maximally slow change" in the data.

A TICA plot visualizes the projection onto the first two TICA basis vectors. To create TICA plots for our data, we first apply the transformation $\phi$ as in term (6) to account for periodicity in our data. In Figures 5 and 6 we display TICA plots for alanine tetrapeptide and chignolin respectively. In both figures the right plot shows the empirical distribution of torsion angles obtained from MD simulations, while the left plot displays the distribution fit of a mixture consisting of 300 components. While for alanine tetrapeptide the learned distribution almost perfectly matches the empirical distribution, we observe small differences for chignolin for some areas of lower probability density (note the logarithmic color scale). Overall we note that both figures indicate that the empirical distributions were well fitted.

## 4 Conclusion

We introduced the shifted inverse stereographic normal distributions (SISND) as a flexible distribution family on the hypertorus, inherently accounting for its topology. We identified unimodality conditions for the proposed distribution by mathematically proving that the density possesses a unique maximum if and

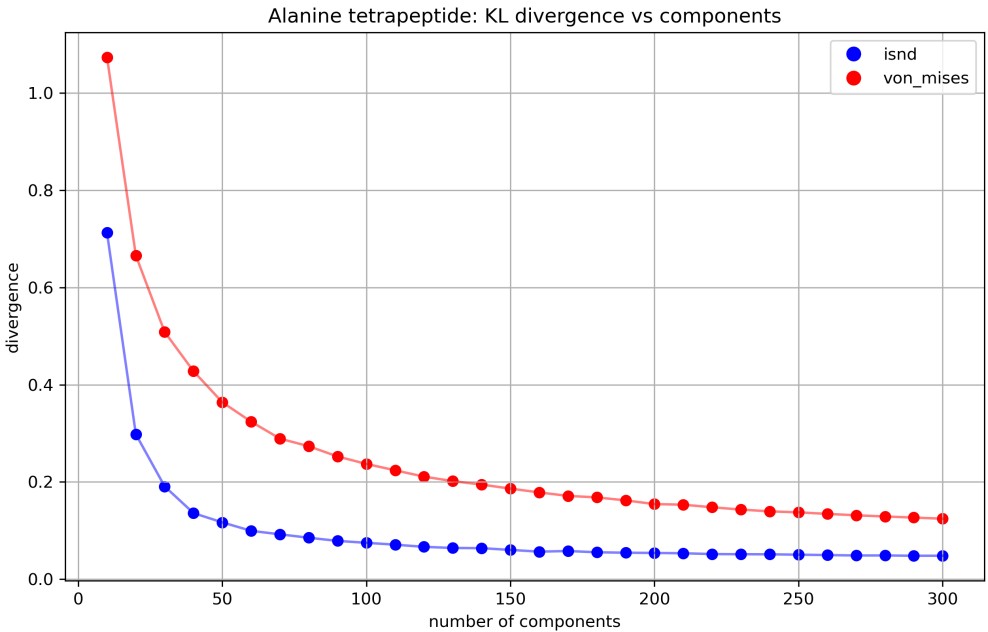

Figure 3: Comparison of the quality of the distribution fit (alanine tetrapeptide backbone torsion angles) of mixtures of ISND vs mixtures of independent von Mises for increasing number of mixture components.

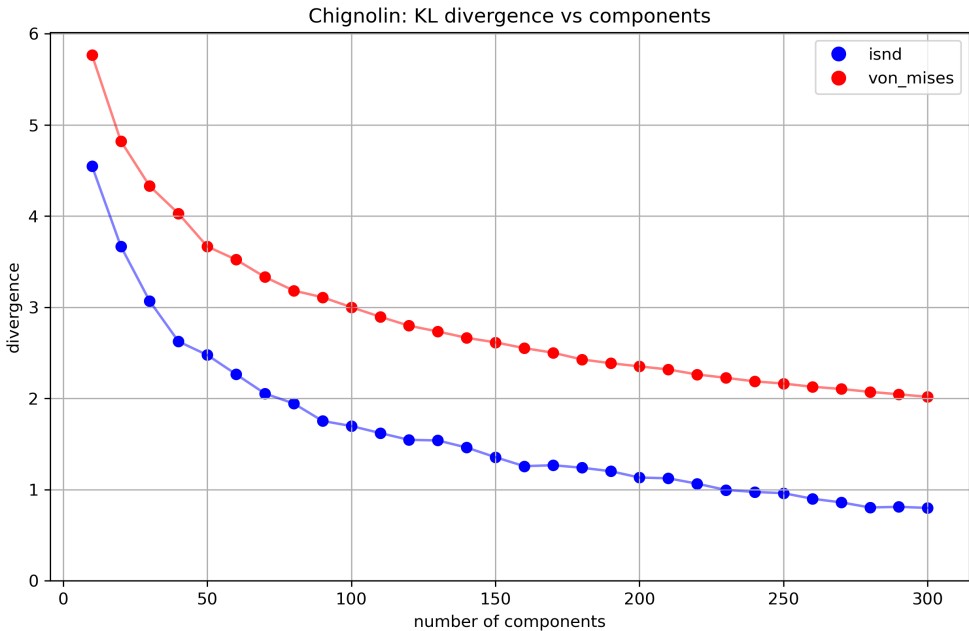

Figure 4: Comparison of the quality of the distribution fit (chignolin backbone torsion angles) of mixtures of isnd vs mixtures of independent von mises for increasing number of mixture components.

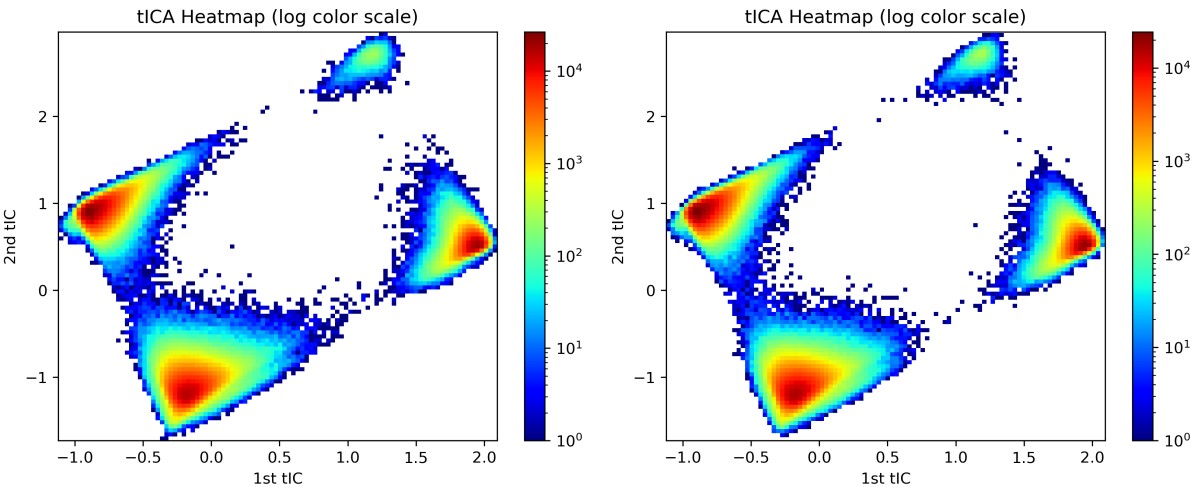

Figure 5: TICA plots for alanine tetrapeptide. Left: Fitted distribution consisting of 300 mixture components. Right: target distribution, obtained from MD simulations.

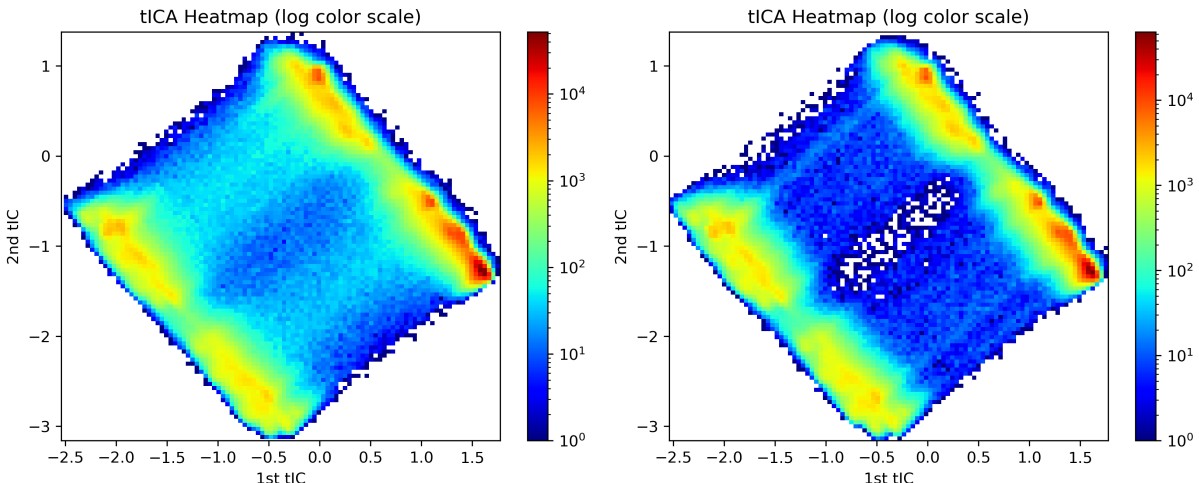

Figure 6: TICA plots for chignolin. Left: Fitted distribution consisting of 300 mixture components. Right: target distribution, obtained from MD simulations.

only if all eigenvalues of the covariance matrix are less than or equal to 0.5. By fitting mixtures of SISND to toroidal data for several examples we demonstrated the applicability of the model. For the example of protein backbones in 6 and 18 dimensions, the fitting performance was verified in terms of the KL-divergence, compared against a baseline model and was visualized in TICA plots. In future work mixtures of shifted inverse stereographic normal distributions might serve as expressive prior distributions for normalizing flow models, enabling a refined learning of densities on the high dimensional hypertorus.

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

## Appendix A  Graphics

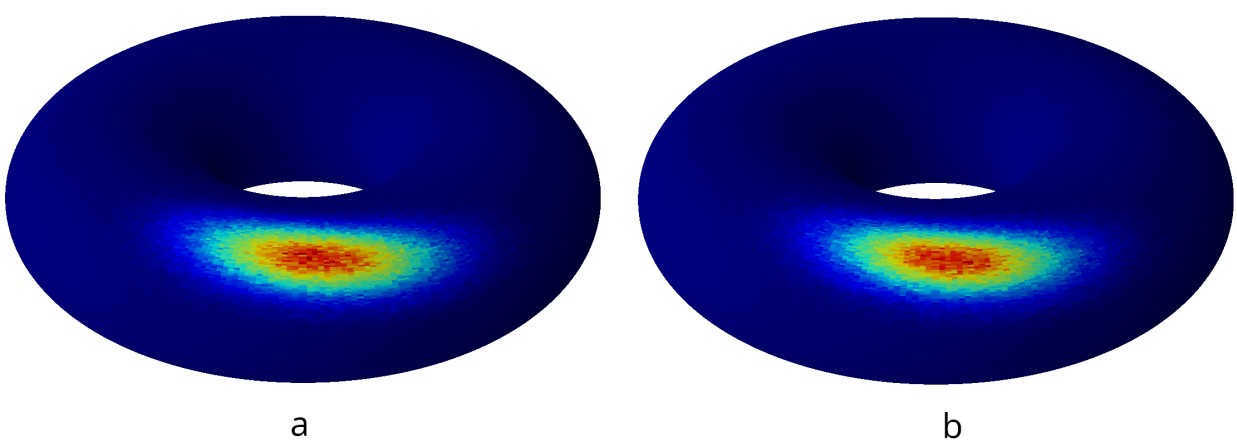

Figure 7: Density visualization on the 2 dimensional torus. One component of a SISND (a) was fitted to a wrapped normal distribution (b). Note that for small eigenvalues, the SISND behaves similarly to the wrapped normal distribution.

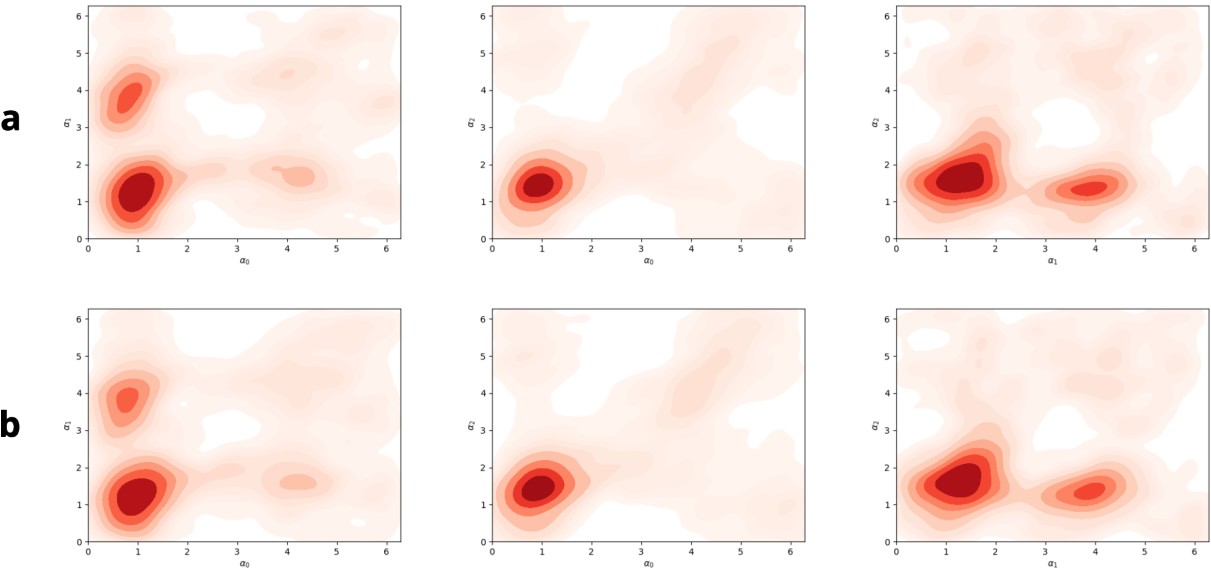

Figure 8: Comparison of marginal distributions of wind direction data. A mixture model of 50 components (b) was fitted to the data distribution (a).

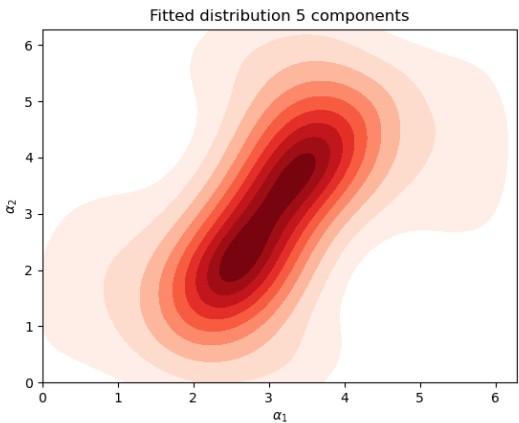 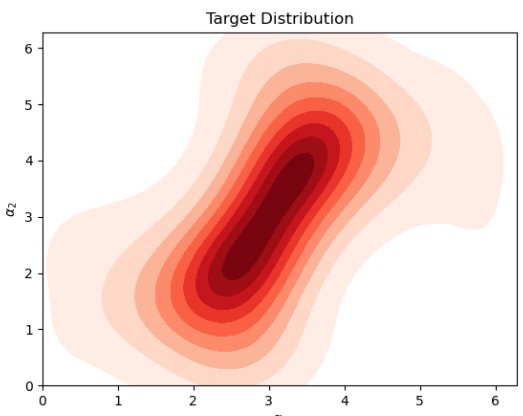

Figure 9: Density visualization of a 5 components mixture SISND fitted to the *Sine model* (see Mardia et al. (2007).

## Appendix B    Notation

Let us agree to the following notation:

**Notation 1.** *Given a subset $U \subset \mathbb{R}$ and a function $g : U \longrightarrow \mathbb{R}$, we denote by*

$$
\begin{aligned}
\underline{g} : U^n &\longrightarrow \mathbb{R}^n \\
(x_1, \cdots, x_n) &\mapsto (g(x_1), \cdots, g(x_n))
\end{aligned}
\tag{8}
$$
$$\tag{9}$$

*the component-wise application of the function $g$ to a vector $(x_1, \ldots, x_n) \in U^n$.*

**Notation 2.** *For $d = (d_1, \ldots, d_n) \in \mathbb{R}^n$ denote by*

$$
\mathrm{Diag}(d) := \begin{pmatrix} d_1 & 0 & \cdots & 0 \\ 0 & d_2 & \cdots & 0 \\ \vdots & \vdots & \ddots & \vdots \\ 0 & 0 & \cdots & d_n \end{pmatrix} \in \mathbb{R}^{n \times n}
$$

*the diagonal matrix of $d$.*

**Notation 3.** *For any hermitian matrix $S \in \mathbb{C}^{n \times n}$ denote by $(\lambda_i(S))_{i=1}^n$ the decreasingly ordered sequence of eigenvalues of $S$ (i.e. $\lambda_1(S) \geq \cdots \geq \lambda_n(S)$).*

## Appendix C    Mixture of independent joint distribution of von Mises distributions

As a baseline in the example 3.3 we use a mixture model of independent joint distribution of von Mises distributions. Formally, in $n \in \mathbb{N}$ dimensions for $\underline{\mu} = (\mu_1, \cdots, \mu_n) \in [-\pi, \pi[^n$ and $\underline{\kappa} = (\kappa_1, \cdots, \kappa_n) \in \mathbb{R}_+$, the density of one such component is defined as

$$
f_{\underline{\mu}, \underline{\kappa}}(\alpha_1, \cdots, \alpha_n) := \left( f_{\mu_1, \kappa_1}(\alpha_1), \cdots, f_{\mu_n, \kappa_n}(\alpha_n) \right),
\tag{10}
$$

where $f_{\mu_i, \kappa_i}(\alpha_i), i \in 1, \cdots, n$ is the von Mises density as defined in (1).

We refer to a mixture model consisting of $m \in \mathbb{N}$ components of the form (10) as *mixture of independent joint distributions of von Mises distributions.*

## Appendix D   Estimation of the Kullback-Leibler divergence

**Definition 2** (**Kullback-Leibler divergence estimator**). *Let $d \in \mathbb{N}$ and $p : \mathbb{R}^d \to \mathbb{R}_+$ be a density on $\mathbb{R}^d$. For $n \in \mathbb{N}$ let*

$$x_1^p, \cdots, x_n^p \overset{i.i.d}{\sim} p$$

*be i.i.d. samples from $p$ and define*

$$\mathcal{X}_p^n := \{x_1^p, \cdots, x_n^p\}$$

*to be the set of samples. For $x \in \mathcal{X}_p^n$, let*

$$d_{\mathcal{X}_p^n \setminus \{x\}}(x) := \min_{y \in \mathcal{X}_p^n \setminus \{x\}} \|x - y\|_2$$

*be the nearest neighbor distance of $x$ in $\mathcal{X}_p^n$. We define an estimator for the Kullback-Leibler divergence as*

$$\hat{D}_{KL}^n (p\|q) := \frac{1}{n} \sum_{x \in \mathcal{X}_p^n} \left( \log \left( \frac{\hat{p}_n(x)}{q(x)} \right) \right) + \Gamma'(1), \tag{11}$$

*where*

$$\hat{p}_n(x) := \frac{\Gamma\left(\frac{d}{2} + 1\right)}{(n-1)\pi^{\frac{d}{2}}} \cdot \frac{1}{(d_{\mathcal{X}_p^n \setminus \{x\}}(x))^d}$$

*and $\Gamma : \mathbb{R}_+ \longrightarrow \mathbb{R}_+$ is the gamma function. Note that the derivative of the gamma function at one, denoted by $\Gamma'(1)$, corresponds to the negative Euler-Mascheroni constant. From Theorem 2 and Corollary 1 in Perez-Cruz (2008) it follows*

$$\hat{D}_{KL}^n (p\|q) \xrightarrow[n\to\infty]{a.s.} D_{KL} (p\|q)$$

*if $p, q$ are absolutely continuous.*

## Appendix E   Lemmas

**Lemma 1.** *Let $f, g : \mathbb{R}^m \longrightarrow \mathbb{R}^n$ be two differentiable functions. Let*

$$\begin{aligned} h : \mathbb{R}^m &\longrightarrow \mathbb{R} \\ x &\mapsto \langle f(x), g(x) \rangle \end{aligned}$$

*be the standard scalar product of $f$ and $g$. Then*

$$D_h(x) = f(x)^T D_g(x) + g(x)^T D_f(x).$$

*Proof.* This can be easily verified by applying the chain rule on partial derivatives of $h$. $\square$

**Lemma 2.** *Let $\Sigma \in S_+^n$. The differential of $f_{\Sigma,\underline{0}}$ (see Definition 1) is given by*

$$D_{f_{\Sigma,\underline{0}}}(\alpha) = f_{\Sigma,\underline{0}}(\alpha) \cdot \underline{\tan}\left(\frac{\alpha}{2}\right)^T \left( I_n - \frac{1}{2}\Sigma^{-1} \operatorname{Diag}\left( \frac{1}{\underline{\cos^2(\frac{\alpha}{2})}} \right) \right)$$

*Proof.* First note that by Lemma 1 we find

$$
\begin{aligned}
\frac{d}{d\alpha}\left(\underline{\tan}\left(\frac{\alpha}{2}\right)^T \Sigma^{-1} \underline{\tan}\left(\frac{\alpha}{2}\right)\right) &= \underline{\tan}\left(\frac{\alpha}{2}\right)^T \Sigma^{-1}\left(\frac{1}{2}\operatorname{Diag}\left(\frac{1}{\cos^2\left(\frac{\alpha}{2}\right)}\right)\right) \\
&+ \left(\Sigma^{-1}\underline{\tan}\left(\frac{\alpha}{2}\right)\right)^T\left(\frac{1}{2}\operatorname{Diag}\left(\frac{1}{\cos^2\left(\frac{\alpha}{2}\right)}\right)\right) \\
&= \underline{\tan}\left(\frac{\alpha}{2}\right)^T \Sigma^{-1}\left(\operatorname{Diag}\left(\frac{1}{\cos^2\left(\frac{\alpha}{2}\right)}\right)\right)
\end{aligned}
\tag{12}
$$

Furthermore, note that

$$
\frac{d}{d\alpha}\left(\prod_{i=1}^{n}\left(\frac{1}{1+\cos(\alpha_i)}\right)\right) = \underline{\tan}\left(\frac{\alpha}{2}\right)^T\prod_{i=1}^{n}\left(\frac{1}{1+\cos(\alpha_i)}\right).
\tag{13}
$$

Using equations (12), (13) and the product rule, we find

$$
\begin{aligned}
D_{f_{\Sigma,\underline{0}}}(\alpha) &= \frac{d}{d\alpha}\left(C\cdot\prod_{i=1}^{n}\left(\frac{1}{1+\cos(\alpha_i)}\right)e^{-\frac{1}{2}\underline{\tan}\left(\frac{\alpha}{2}\right)^T\Sigma^{-1}\underline{\tan}\left(\frac{\alpha}{2}\right)}\right) \\
&= C\frac{d}{d\alpha}\left(\prod_{i=1}^{n}\left(\frac{1}{1+\cos(\alpha_i)}\right)\right)e^{-\frac{1}{2}\underline{\tan}\left(\frac{\alpha}{2}\right)^T\Sigma^{-1}\underline{\tan}\left(\frac{\alpha}{2}\right)} \\
&+ C\prod_{i=1}^{n}\left(\frac{1}{1+\cos(\alpha_i)}\right)\frac{d}{d\alpha}e^{-\frac{1}{2}\underline{\tan}\left(\frac{\alpha}{2}\right)^T\Sigma^{-1}\underline{\tan}\left(\frac{\alpha}{2}\right)} \\
&= C\prod_{i=1}^{n}\left(\frac{1}{1+\cos(\alpha_i)}\right)e^{-\frac{1}{2}\underline{\tan}\left(\frac{\alpha}{2}\right)^T\Sigma^{-1}\underline{\tan}\left(\frac{\alpha}{2}\right)}\cdot\underline{\tan}\left(\frac{\alpha}{2}\right)^T \\
&\cdot\left(I_n - \frac{1}{2}\Sigma^{-1}\operatorname{Diag}\left(\frac{1}{\cos^2\left(\frac{\alpha}{2}\right)}\right)\right) \\
&= f_{\Sigma,0}(\alpha)\cdot\underline{\tan}\left(\frac{\alpha}{2}\right)^T\left(I_n - \frac{1}{2}\Sigma^{-1}\operatorname{Diag}\left(\frac{1}{\cos^2\left(\frac{\alpha}{2}\right)}\right)\right),
\end{aligned}
$$

where

$$
C = (2\pi)^{-\frac{n}{2}}\det(\Sigma)^{-\frac{1}{2}}.
$$

$\square$

**Lemma 3.** *The Hessian matrix of $f_{\Sigma,\underline{0}}$ at $\alpha = \underline{0}$ is given by*

$$
H_{f_{\Sigma,\underline{0}}}(\underline{0}) = (2\pi)^{-\frac{n}{2}}\det(\Sigma)^{-\frac{1}{2}}\left(\frac{1}{2}\right)^{n+1}\cdot\left(I_n - \frac{1}{2}\Sigma^{-1}\right)
\tag{14}
$$

*Proof.* For the sake of clarity, let us define

$$
h(\alpha) := \left(I_n - \frac{1}{2}\operatorname{Diag}\left(\frac{1}{\cos^2\left(\frac{\alpha}{2}\right)}\right)\Sigma^{-1}\right)\underline{\tan}\left(\frac{\alpha}{2}\right),
$$

hence

$$
\operatorname{grad}_{f_{\Sigma,\underline{0}}}(\alpha) = f_{\Sigma,\underline{0}}(\alpha)\cdot h(\alpha)
$$

The Hessian matrix is the differential of the gradient of $f_{\Sigma,\underline{0}}$

$$
\begin{aligned}
H_{f_{\Sigma,\underline{0}}}(\underline{0}) \quad &= \quad D_{\mathrm{grad}_{f_{\Sigma,\underline{0}}}}(\underline{0}) \\
&= \quad f_{\Sigma,0}(\underline{0}) \cdot D_h(\underline{0}) + D_{f_{\Sigma,0}}(\underline{0}) \otimes h(\underline{0}) \\
&\underset{h(\underline{0})=\underline{0}}{=} \quad f_{\Sigma,0}(\underline{0}) \cdot D_h(\underline{0}),
\end{aligned}
$$

where $\otimes$ denotes the Kronecker product. Applying Lemma 1 component-wise and observing that $\underline{\tan}(\underline{0}) = 0$, we find

$$
\begin{aligned}
D_h(\underline{0}) \quad &= \quad \left( I_n - \frac{1}{2} \mathrm{Diag} \left( \frac{1}{\underline{\cos}^2(\underline{0})} \right) \Sigma^{-1} \right) \frac{d}{d\alpha} \underline{\tan}(\alpha) \Big|_{\alpha=\underline{0}} \\
&= \quad \left( I_n - \frac{1}{2} \Sigma^{-1} \right) \frac{1}{2} \mathrm{Diag} \left( \frac{1}{\underline{\cos}^2(\frac{\alpha}{2})} \right) \Big|_{\alpha=\underline{0}} \\
&= \quad \frac{1}{2} \left( I_n - \frac{1}{2} \Sigma^{-1} \right)
\end{aligned}
\tag{15}
$$

Moreover it holds

$$
f_{\Sigma,\underline{0}}(\underline{0}) = (2\pi)^{-\frac{n}{2}} \det(\Sigma)^{-\frac{1}{2}} \cdot \left( \frac{1}{2} \right)^n .
\tag{16}
$$

Plugging equations (15),(16) into (14) completes the proof.

$\square$

**Lemma 4** (Weyl inequality). *Let $n \in \mathbb{N}$ and for any hermitian matrix $S \in \mathbb{C}^{n \times n}$ denote by $(\lambda_i(S))_{i=1}^n$ the decreasingly ordered eigenvalues of $S$ (i.e. $\lambda_1(S) \geq \cdots \geq \lambda_n(S)$). Let $A, B \in \mathbb{C}^{n \times n}$ be Hermitian matrices. Then for any $i, j \in \{1, \ldots, n\}$ with $i + j \leq n + 1$ it holds*

$$
\lambda_{i+j-1}(A + B) \leq \lambda_i(A) + \lambda_j(B).
\tag{17}
$$

## Appendix F   Proof of main theorem

**Theorem 1.** *Let*

$$
\mathcal{A} := \left\{ \Sigma \in S_+^n \,|\, \lambda_{max}(\Sigma) \leq 0.5 \right\}
\tag{5}
$$

*be the set of SPD matrices with all eigenvalues being less or equal than 0.5. For $\Sigma \in S_+^n, \mu \in [-\pi, \pi[^n$ let $f_{\Sigma,\mu}$ be defined as in Definition 1. Then $f_{\Sigma,\mu}$ is unimodal if and only if $\Sigma \in \mathcal{A}$.*

*Proof.* Obviously, the parameter $\mu \in [-\pi, \pi[^n$ only shifts the density function, s.th. the number of modes only depends on $\Sigma \in S_+^n$ and we may assume $\mu = 0$. First, we observe that $f_{\Sigma,\underline{0}}$ is point-symmetric around $\underline{0} \in \mathbb{R}^n$:

$$
\forall \alpha \in [-\pi, \pi[^n : \quad f_{\Sigma,\underline{0}}(\alpha) = f_{\Sigma,\underline{0}}(-\alpha).
\tag{18}
$$

Hence, the function $f_{\Sigma,\underline{0}}(\alpha)$ is unimodal if and only if $\underline{0} \in [-\pi, \pi[^n$ is a maximum and is the unique maximum.

Let us first assume $\Sigma \in \mathcal{A}$. Note that $\underline{0} \in [-\pi, \pi[^n$ being the unique maximum of $f_{\Sigma,\underline{0}}(\alpha)$ is equivalent to

$$
\begin{aligned}
D_{f_{\Sigma,\underline{0}}}(\alpha) \quad &= \quad 0, \quad && \text{if } \alpha = 0 \tag{19} \\
D_{f_{\Sigma,\underline{0}}}(\alpha) \quad &\neq \quad 0, \quad && \text{if } \alpha \in \,]-\pi, \pi[^n \setminus \{0\}, \tag{20}
\end{aligned}
$$

since $f_{\Sigma,\underline{0}}(\alpha)$ is differentiable in $]-\pi,\pi[^n$ and does not take its maximum at $\alpha = -\pi$. By Lemma 2, the differential is given by

$$
\begin{aligned}
D_{f_{\Sigma,\underline{0}}}(\alpha) &= f_{\Sigma,0}(\alpha) \cdot \underline{\tan}\left(\frac{\alpha}{2}\right)^T \left(I_n - \frac{1}{2}\Sigma^{-1}\operatorname{Diag}\left(\frac{1}{\cos^2(\frac{\alpha}{2})}\right)\right) \\
&= f_{\Sigma,0}(\alpha) \cdot \underline{\tan}\left(\frac{\alpha}{2}\right)^T \underbrace{\left(\operatorname{Diag}\left(\underline{\cos}^2\left(\frac{\alpha}{2}\right)\right) - \frac{1}{2}\Sigma^{-1}\right)}_{A(\alpha):=} \operatorname{Diag}\left(\frac{1}{\cos^2(\frac{\alpha}{2})}\right)
\end{aligned}
$$

The gradient (i.e. the transposed differential) is thus given by

$$
\operatorname{grad}_{f_{\Sigma,\underline{0}}}(\alpha) = f_{\Sigma,0}(\alpha) \cdot \operatorname{Diag}\left(\frac{1}{\cos^2\left(\frac{\alpha}{2}\right)}\right) A(\alpha) \cdot \underline{\tan}\left(\frac{\alpha}{2}\right). \tag{21}
$$

Since $\underline{\tan}(\underline{0}) = \underline{0}$, we find $\operatorname{grad}_{f_{\Sigma,\underline{0}}}(\underline{0}) = \underline{0}$ which shows (19). To show (20), first observe that (21) implies

$$
\forall \alpha \in ]-\pi,\pi[^n \setminus \{0\}: \qquad \operatorname{grad}_{f_{\Sigma,\underline{0}}}(\alpha) = \underline{0} \quad \implies \quad \det(A(\alpha)) = 0. \tag{22}
$$

We will now use Weyl's inequality (Lemma 4, for a proof see for example section 5.1 of Helmke & Rosenthal (1995)) on eigenvalues to show $\lambda_1(A(\alpha)) < 0$ which results in $\det(A(\alpha)) \neq \underline{0}$ and thus by (22) shows $\operatorname{grad}_{f_{\Sigma,\underline{0}}}(\alpha) \neq \underline{0}$.

By equation (17), we find

$$
\lambda_1(A(\alpha)) \leq \lambda_1\left(\operatorname{Diag}\left(\underline{\cos}^2\left(\frac{\alpha}{2}\right)\right)\right) + \lambda_1\left(-\frac{1}{2}\Sigma^{-1}\right). \tag{23}
$$

Since we assumed $\alpha \in ]-\pi,\pi[^n \setminus \{0\}$, it holds

$$
\lambda_1\left(\operatorname{Diag}\left(\underline{\cos}^2\left(\frac{\alpha}{2}\right)\right)\right) < 1. \tag{24}
$$

Furthermore $\Sigma \in \mathcal{A}$ implies

$$
\lambda_1\left(-\frac{1}{2}\Sigma^{-1}\right) = -\frac{1}{2}\lambda_n\left(\Sigma^{-1}\right) = -\frac{1}{2 \cdot \lambda_1(\Sigma)} \leq -1. \tag{25}
$$

Combining equations (23),(24),(25) we deduce

$$
\lambda_1(A(\alpha)) < 0,
$$

which proves $\det(A(\alpha)) \neq \underline{0}$ and hence by (22) we find $\operatorname{grad}_{f_{\Sigma,\underline{0}}}(\alpha) \neq \underline{0}$.

We now show that $f_{\Sigma,\underline{0}}$ is not unimodal for $\Sigma \notin \mathcal{A}$. By Lemma 3, the Hessian matrix of $f_{\Sigma,\underline{0}}$ is given by

$$
H_{f_{\Sigma,\underline{0}}}(\underline{0}) = (2\pi)^{-\frac{n}{2}} \det(\Sigma)^{-\frac{1}{2}} \left(\frac{1}{2}\right)^{n+1} \cdot \left(I_n - \frac{1}{2}\Sigma^{-1}\right).
$$

Hence

$$
\begin{aligned}
\lambda_1\left(H_{f_{\Sigma,\underline{0}}}(\underline{0})\right) &= (2\pi)^{-\frac{n}{2}} \det(\Sigma)^{-\frac{1}{2}} \left(\frac{1}{2}\right)^{n+1} \left(1 - \frac{1}{2}\lambda_n\left(\Sigma^{-1}\right)\right) \\
&= (2\pi)^{-\frac{n}{2}} \det(\Sigma)^{-\frac{1}{2}} \left(\frac{1}{2}\right)^{n+1} \left(1 - \frac{1}{2}\frac{1}{\lambda_1(\Sigma)}\right).
\end{aligned}
$$

We conclude

$$\forall i \in \{1, \cdots, n\}: \quad \lambda_i\left(H_{f_{\Sigma,\underline{0}}}(0)\right) \leq 0 \quad \Longleftrightarrow \quad \lambda_1\left(H_{f_{\Sigma,\underline{0}}}(0)\right) \leq 0$$

$$\Longleftrightarrow \quad \lambda_1\left(\Sigma\right) \leq \frac{1}{2}$$

$$\Longleftrightarrow \quad \forall i \in \{1, \cdots, n\}: \quad \lambda_i\left(\Sigma\right) \leq \frac{1}{2}.$$

Thus, $\Sigma \notin \mathcal{A}$ implies that $\alpha = \underline{0}$ is not a maximum of $f_{\Sigma,\underline{0}}$, which by the symmetry argument in the context of (18) shows multimodality. $\qquad \square$

