# OpenReview forum: "Shifted Inverse stereographic normal distributions as flexible distribution family on the hypertorus"
_TMLR — Rejected by TMLR_

### Review · Reviewer_t7YF · 2024-02-09

**Summary Of Contributions:**

The paper considers a family of distributions defined on the hypertorus (i.e., the product of $n$ copies of the circle), obtained from Gaussian distributions on $\mathbb{R}^n$ by pushforward via coordinate-wise inverse stereographic projection. This distribution family, termed as shifted inverse stereographic normal distributions, is presented as a computationally efficient and flexible alternative for modeling circular data.

**Audience:**

Yes

**Broader Impact Concerns:**

None.

**Claims And Evidence:**

No

**Requested Changes:**

The **most important requested change** is to provide a detailed comparison with the work of Selvitella (2019), explicitly highlighting any novel contributions of the current paper, if present.

Here are some further comments and typos:
- In the title, "a" should precede "flexible".
- Spacing: please use \DeclarePairedDelimiter to correct the spacing issues for $]-\pi, \pi[$.
- Pg. 3, this is the pushforward of the Gaussian under $h_n^{-1}$ (not $h_n$).
- Thm. 1, the more standard convention is that the eigenvalues are sorted in decreasing order.
- Pg. 5, "Since multimodality of $f_{\Sigma,\mu}$ might cause many local minima": is there justification for this claim?
- Sec. 3, "sinkhorn" should be capitalized.
- Below eq. (18), "it's" should be "its".
- Please use $\implies$ and $\iff$ instead of $=>$ and $<=>$.

**Strengths And Weaknesses:**

I would like to bring to everyone’s attention that **this work shares substantial overlap with the cited work of Selvitella (2019)**. This previous work has already defined this family of distributions, and even the unimodality result presented in this work seems to already be present in the literature. Although TMLR does not list originality as an acceptance criterion, I do not feel that we can accept **plagiarism**.

Regarding the content itself: the experiment is promising, but I also do not feel that this idea has been sufficiently explored and tested on enough examples to conclude that the claims made in this paper are substantiated by evidence.

---

> ### Author Response · Authors · 2024-03-12
>
> Dear reviewer t7YF,
>
> Thank you very much for your feedback.
>
> We can assure you that it was not our intention to claim the scientific contributions of  Selvitella (2019) as our own. Although we cited the work, the distinction between our contribution and the existing literature might not have been clear enough. We now added two paragraphs clearly distinguishing and clarifying the contribution of our work compared to Selvitella (2019). We changed:
>
> "In this work we present the shifted inverse stereographic normal distribution as an appealing alternative,
> having a tractable density and its mixtures being flexible enough to capture non-trivial toroidal distributions
> in higher dimensions. Many statistical properties of the inverse stereographic projection were discussed in
> Selvitella (2019), mostly for the one dimensional case."
>
> to
>
> "
> Selvitella (2019) suggested the inverse stereographic projection of normal distributions, termed inverse stere-
> ographic normal distributions, as a further flexible alternative to the von Mises distribution. Beyond having
> an easy tractable density in the multivariate case, the distribution was shown in Selvitella (2019) to have
> many favorable statistical properties. Specifically, the inverse stereographic normal distribution is closed
> under marginalization and conditioning, has asymptotic relations to the von Mises and wrapped normal
> distributions and is the limit distribution in a toroidal analogue of the central limit theorem. Furthermore
> Selvitella (2019) stated unimodality conditions for the inverse stereographic normal distributions in the one
> dimensional case and demonstrated applications in one and two dimensions.
>
> In this work we consider shifted versions of the inverse stereographic projection of zero centered normal
> distributions, which we term shifted inverse stereographic normal distributions in reference to Selvitella
> (2019). We demonstrate the flexibility and practical applicability of the distribution family by fitting mixtures
> of shifted inverse stereographic normal distributions to non-trivial toroidal distributions in higher dimensions.
> Furthermore we generalize the unimodality result of Selvitella (2019) for the mean-free case to arbitrary
> dimensions.
> "
>
> Furthermore we fixed the mistakes and style/convention issues you mentioned.
> The sentence
> "Since multimodality of $f_{\Sigma,\mu}$ might cause many local minima in the training objective function, we restrict learning to the set $\mathcal{A}$"
>
> was changed to
> "Empirically we found it beneficial to restrict the learning to the set of unimodality $\mathcal{A}$ "
>
>
> Within the next few days we would like to add further examples, we apologize for the delay.

---

> > ### Author Response · Authors · 2024-03-18
> >
> > Dear reviewer t7YF,
> >
> > Thank you very much for your patience!
> >
> > As you suggested, we further explored the distribution: We added further examples, included more graphics and compared against a base line.
> > In more detail the following additions were made:
> >
> > * As an extra real-world example we included trivariate wind direction data. The Figure 8 visualizes the target distribution vs the fitted distribution.
> > * A visualization (Figure 7) was added, showing a comparison of ISND to wrapped normal, demonstrating the similarity for small eigenvalues
> > * As a synthetic example, we included the Sine model (special case of the bivariate von Mises distribution) and fitted it with our model.
> > * The performance evaluation on alanine tetrapeptide and chignolin now contains a comparison against a baseline model (independent joint distribution of von Mises distributions).
> > * Also a more complex distribution was used for chignolin (it is more exhaustively sampled). The details of the data generation (parameters and algorithmic details) are provided by the reference cited in our work.

---

### Review · Reviewer_B121 · 2024-02-17

**Summary Of Contributions:**

In this paper, the authors propose a new family of probability distributions defined over the hypertorus that is obtained by pushing forward a standard Gaussian measure on $\mathbb{R}^n$ via stereographic projection.

**Audience:**

Yes

**Broader Impact Concerns:**

There are no immediate ethical implications of the work.

**Claims And Evidence:**

No

**Requested Changes:**

I believe the work will be strengthened by having a more thorough investigation of the model by comparing against some baseline. This can even be a very simple model like having n-copies of 1D toroidal distributions. There is also this work (https://arxiv.org/pdf/1602.05003.pdf) that defines von-Mises distribution over the hypertorus, which seems relevant and possibly a model that the authors can compare against. Also, I would like to ask the authors to please check (11) as I am not seeing how that is true. This calls into question whether the result in Theorem 1 is true.

**Strengths And Weaknesses:**

Strength:
- The paper is written clearly.
- The motivation for introducing the model is sound. It addresses the problem of intractability of existing models on the torus in high dimensions.

Weaknesses:
- Is equation (11) in the appendix true? Instead of $\tan(\alpha)$, it seems to me that you will get $\sin(\alpha) / (1 + \cos(\alpha))$.
- There are no baseline comparisons in the experiments, which makes it difficult to fully judge the benefits of SISND. For example, you can demonstrate the complexity of the wrapped normal distribution with increasing dimension (perhaps on a toy dataset) to show that SISND does not have such scaling issue. You could have also used for example a simple baseline of $n$-independent von Mises distributions in the experiment to have something to compare against. At the moment, the claims in the paper are not fully justified by the experiments due to a lack of comparisons (e.g. the authors write "we note that both figures indicate that the empirical distributions were well fitted", but compared to what? Also how do we determine whether the results in figures 3 and 4 are good without a baseline?).

---

> ### Author Response · Authors · 2024-03-12
>
> Dear reviewer B121,
>
> Thank you very much for your feedback.
>
> * Equation (11) is justified by the trigonometric equality of \sin(\alpha)/(1+\cos(\alpha))= \tan(\alpha/2). As a reference see e.g.
> here https://en.wikipedia.org/wiki/Tangent_half-angle_formula.
> * As you suggested, we implemented a base line comparison against the independent von Mises distribution. We aim to deliver the results for that and some further examples within the next few day.
> We apologize for the delay...

---

> > ### Author Response · Authors · 2024-03-18
> >
> > Dear reviewer B121,
> >
> > Thank you very much for your patience!
> >
> > We now included several further examples and the baseline comparison that you suggested (n-copies of 1D toroidal distributions).
> > In more detail, the following additions were made:
> >
> > * As an extra real-world example we included trivariate wind direction data. The Figure 8 visualizes the target distribution vs the fitted distribution.
> > * A visualization (Figure 7) was added, showing a comparison of ISND to wrapped normal, demonstrating the similarity for small eigenvalues
> > * As a synthetic example, we included the Sine model (special case of the bivariate von Mises distribution) and fitted it with our model.
> > * The performance evaluation on alanine tetrapeptide and chignolin now contains a comparison against the baseline model.
> > * Also a more complex distribution was used for chignolin (it is more exhaustively sampled). The details of the data generation (parameters and algorithmic details) are provided by the reference cited in our work.

---

### Review · Reviewer_u67e · 2024-02-27

**Summary Of Contributions:**

The authors propose a normal-like distribution on the circle with a tractable normalization constant relying on a simple trick, which is extended to model distributions on hyper-toruses under a mixture model scheme. The efficiency of the new statistical model is demonstrated on simulated protein data.

**Audience:**

Yes

**Broader Impact Concerns:**

I believe there are no direct ethical implications.

**Claims And Evidence:**

No

**Requested Changes:**

Please see the weaknesses.

**Strengths And Weaknesses:**

Strengths:
- I find the main idea interesting as the distribution is rather simple, yet effective.
- The theoretical result regarding multimodality seems to be reasonable.
- The proposed model is effective based on the included demonstrations.

Weaknesses:
- I think that some figures can be included to help the exposition of the idea. For example, distributions on a 2-dimensional torus in a 3-dimensional space can be included as a synthetic experiment.
- The writing is in general fine, but in some parts it can be improved. For example, I think that Eq. 2 can be described better, as well as Definition 1. Similarly, the experimental section is rather concise, and some further details can be included in the appendix.
- I believe that some additional experiments are necessary. For example, synthetic complex distributions on 2-dim torus in 3-dim, comparisons with the related methods, extra real-world problems (e.g. directional data).

---

> ### Author Response · Authors · 2024-03-12
>
> Dear reviewer u67e,
>
> Thank you very much for your feedback.
>
> We improved the description of equation (2) by adding the explaining sentence:
> "In other words, mapping a N(0,Σ) distributed random variable to ]−π, π[ via h^{−1}_{n}
>  results in a random variable distributed according to (2)"
>
> The relation of equation (2) to Definition 1 was clarified by setting a new paragraph and changing
>
> "Note that this density can be continuously extended to [−π, π[^{n}. Furthermore it can be shifted around the torus by
> subtraction of a center μ ∈ [−π, π[^{n} from the function argument. Formally, we define the shifted inverse
> stereographic normal density (SISND) as follows:"
>
> to
>
> "We now define the shifted inverse stereographic normal density (SISND) by continuously extending the density
> (2) to [−π, π[^{n} and subtracting a shifting parameter in its argument."
>
>
> As for further experiments we apologize for the delay and kindly ask for a bit more time.
> We will deliver those timely.

---

> > ### Author Response · Authors · 2024-03-18
> >
> > Dear reviewer u67e,
> >
> > Thank you very much for your patience!
> >
> > We now included several further experiments and provided a base line comparison. In detail:
> > * As you suggested we included a visualization of the distribution on the 2D torus. The Figure 7 visualizes a comparison of ISND vs wrapped normal, demonstrating the similarity for small eigenvalues.
> > * As an extra real-world example we included trivariate wind direction data. The Figure 8 visualizes the target distribution vs the fitted distribution.
> > * We also included the Sine model (special case of the bivariate von Mises distribution) as a synthetic example.
> > * As you suggested, we included a comparison to a baseline model (mixture of independent joint distribution of von Mises distributions)
> > * By including the further examples, and describing details about the baseline model in the appendix, the experiments section was now expanded. A more complex distribution for chignolin was used in the revision (by more exhaustive sampling). The details of the data generation (parameters and algorithmic details) are provided by the reference cited in our work.

---

### Decision · Action_Editor_LwHR · 2024-04-05

**Recommendation:** Reject

**Comment:**

The reviewers generally agreed that the overlap with Selvitella (2019) was too substantial to warrant publication in TMLR.

**Audience:**

The audience for this paper would be limited, given the existing contributions of Selvitella (2019). The reviewers generally agreed that the contribution of the work in view of past work was not interesting enough to warrant publication in TMLR.

**Claims And Evidence:**

This paper introduces a shifted version of the inverse stereographic normal distribution of Selvitella (2019). The paper proves that for a particular subset of parameters, the distribution is unimodal, and shows how to fit mixtures using gradient descent. The paper compares the distribution to the von Mises distribution empirically.

It is not clear what the main claim of the paper is. The paper suggests that the shifted inverse stereographic normal is "flexible and practical" (p2), and "interesting an computationally appealing" (abstract), but does not justify these claims with evidence beyond what the work of Selvitella already achieves.